# Dendritic Cells or Macrophages? The Microenvironment of Human Clear Cell Renal Cell Carcinoma Imprints a Mosaic Myeloid Subtype Associated with Patient Survival

**DOI:** 10.3390/cells11203289

**Published:** 2022-10-19

**Authors:** Dorothee Brech, Anna S. Herbstritt, Sarah Diederich, Tobias Straub, Evangelos Kokolakis, Martin Irmler, Johannes Beckers, Florian A. Büttner, Elke Schaeffeler, Stefan Winter, Matthias Schwab, Peter J. Nelson, Elfriede Noessner

**Affiliations:** 1Immunoanalytics/Tissue Control of Immunocytes, Helmholtz Zentrum München, 81377 Munich, Germany; 2Bioinformatics Core Unit, Biomedical Center, Ludwig-Maximilians-University, 82152 Planegg, Germany; 3Institute of Experimental Genetics, Helmholtz Zentrum München, 85764 Neuherberg, Germany; 4German Center for Diabetes Research (DZD), 85764 Neuherberg, Germany; 5Chair of Experimental Genetics, Technical University of Munich, 85354 Freising, Germany; 6Margarete Fischer-Bosch-Institute of Clinical Pharmacology, 70376 Stuttgart, Germany; 7University of Tuebingen, 72074 Tuebingen, Germany; 8Department of Clinical Pharmacology, University of Tuebingen, 72074 Tuebingen, Germany; 9Department of Pharmacy and Biochemistry, University of Tuebingen, 72074 Tuebingen, Germany; 10German Cancer Consortium (DKTK), Partner Site Tuebingen, German Cancer Research Center (DKFZ), 69120 Heidelberg, Germany; 11Medizinische Klinik und Poliklinik IV, University of Munich, 80336 Munich, Germany

**Keywords:** mononuclear phagocyte system, macrophage plasticity, tissue macrophage, tumor microenvironment, gene expression, VSIG4, NRP1, GPNMB

## Abstract

Since their initial description by Elie Metchnikoff, phagocytes have sparked interest in a variety of biologic disciplines. These important cells perform central functions in tissue repair and immune activation as well as tolerance. Myeloid cells can be immunoinhibitory, particularly in the tumor microenvironment, where their presence is generally associated with poor patient prognosis. These cells are highly adaptable and plastic, and can be modulated to perform desired functions such as antitumor activity, if key programming molecules can be identified. Human clear cell renal cell carcinoma (ccRCC) is considered immunogenic; yet checkpoint blockades that target T cell dysfunction have shown limited clinical efficacy, suggesting additional layers of immunoinhibition. We previously described “enriched-in-renal cell carcinoma” (erc) DCs that were often found in tight contact with dysfunctional T cells. Using transcriptional profiling and flow cytometry, we describe here that ercDCs represent a mosaic cell type within the macrophage continuum co-expressing M1 and M2 markers. The polarization state reflects tissue-specific signals that are characteristic of RCC and renal tissue homeostasis. ErcDCs are tissue-resident with increasing prevalence related to tumor grade. Accordingly, a high ercDC score predicted poor patient survival. Within the profile, therapeutic targets (VSIG4, NRP1, GPNMB) were identified with promise to improve immunotherapy.

## 1. Introduction

Elie Metchnikoff first formulated his phagocyte theory in 1882. While his family visited a circus, he observed with his microscope so-called “mobile cells” in transparent star-fish larva [1,2]. He was studying these cells in the context of digestive processes and on that day, the idea crossed his mind that these cells might also eat microbes. He then performed a simple experiment piercing a thorn into the transparent starfish. The next day, the thorns were surrounded by mobile cells. He concluded that these cells take up and digest invasive bacteria and, thus, considered his theory confirmed. Over the following years, Metchnikoff continued studying “his” mobile cells, which were eventually named phagocytes, a term still in use today [1,2].

His first article on the phagocyte hypothesis was published in 1883. In this report, he provided evidence that mobile cells from frogs were involved in host defense as well as the elimination of dying or degenerating endogenous cells [3]. In 1887, he proposed the idea that macrophages and microphages were capable of phagocytosis [3]. From 1888 to 1916, Mechnikoff conducted his research in Paris at the Pasteur institute [2,4]. It was during this time that he established his theory of innate immunity and, together with Paul Ehrlich, received the Nobel Price “in recognition of their work on immunity” in 1908 [2,3,4]. Since their initial description, much has been learned about the biology of phagocytes, but many mysteries have yet to be explored. While initially thought to be one cell type, a plethora of variations have been identified that are summarized by the concept of the mononuclear phagocyte system (MPS) [5,6,7,8,9,10,11,12]. By responding to innate signals and lymphocyte mediators, phagocytes act as integral components of the immune response to microbes, tissue injury and cell transformation. They play a central role in the control of tissue homeostasis, wound healing and damage prevention. These processes are mediated in part through their capacity to remove cellular debris and induce angiogenesis, as well as by their ability to activate or tolerize relevant immunocytes. The effector activities of MPS cells are strongly dependent on the cell’s polarization state, imprinted by signals from their local environment [13,14,15,16].

Clear cell renal cell carcinoma (ccRCC) arises from the epithelial cells of the renal tubulointerstitium. This tumor type is generally richly infiltrated by immune cells, including T, NK and myeloid cells. However, despite an abundance of CD8^+^ T cells that can recognize and destroy tumor cells when taken out of the ccRCC environment [17,18,19], control of tumor progression fails, suggesting local suppression of T cell effector activity [20,21,22]. Invigorating the T cell response through a blockade of the immune checkpoint molecules PD-1 or PD-L1, in combination with a blockade of the checkpoint CTLA-4 or with tyrosine kinase inhibition (TKI) has been approved as a first line treatment for advanced and metastatic RCC [23,24]. Despite some improvements in management of the disease through these T cell targeting therapies, only a fraction of patients has been shown to respond [23,24,25,26], suggesting that mechanisms beyond those directly targeting the T cells control the antitumor response. This underscores the critical need to better understand the tumor environment and to identify additional therapeutic targets beyond CTLA-4, PD-1 and PD-L1 to expand the range of patients that can be effectively treated.

We have previously reported an unusual myeloid cell type in the ccRCC tissue [27] that co-expresses the macrophage marker CD14 and several DC markers, such as CD209, a marker of interstitial DCs with cross-presentation ability [28], as well as the costimulatory molecules HLA-DR and CD40. We designated this myeloid subpopulation “enriched-in-renal-carcinoma DCs” (ercDCs), due to its strong enrichment in the tumor center, where they represent over 60% of the CD209^+^ population (mean, 62%; range, 26% to 80%), compared to non-tumor kidney cortex (mean, 19%; range, 0% to 43%). In ccRCC tissues, ercDCs have often been found to be tightly engaged with T cells, suggesting intercellular communication in situ [27].

In murine kidney interstitium, DCs with dual macrophage (F4/80) and DC (CD11c) markers have been described as conveying tolerogenic and tissue protective functions [29,30]. Considering this marker analogy, and the observed ercDC/T cell contacts, we hypothesized a functional analogy whereby ercDCs could contribute to tumor immune evasion by shielding emerging tumor cells from immune attack.

If such a scenario is correct, ercDCs could represent a promising therapeutic target if molecules or pathways underlying their functional modulation could be identified. Previous experiments suggested that ercDC were able to cross-present antigen for T cell recognition. However, they did not secrete IL-12 and were unable to perform allogeneic T cell priming in vitro. Further evidence of immunomodulation was seen in co-culture with tumor cells and T cells where the presence of ercDCs enhanced tumor cell proliferation and reduced CD8 T cell recruitment. Immune histology supported this evidence as lower numbers of CD8^+^ and NK cells have been seen in ccRCC tissues with high ercDC content [27].

To help clarify the functional attributes of ercDCs and to identify markers for targeted approaches, as well as to position them within the MPS continuum, the CD14^+^CD209^+^ cell population was sorted from ccRCC tissue by flow cytometry and subjected to transcriptomic profiling. Transcript profiles have been useful in defining subsets and polarization states within the MPS continuum [6] and helped assign functional characteristics as demonstrated, e.g., by Houser et al. who described two distinct subsets of CD14^+^ decidual macrophages, CD11c^HI^ and CD11c^LO^, with distinct functions in tissue remodeling, growth and development [31].

In the present study, we provide a molecular characterization of ercDCs isolated from ccRCC tissues. A definition of functional characteristics as well as their relationship to myeloid subtypes from other human tissues was established. The transcriptomic profiling identified ercDCs as a unique myeloid subset within the macrophage spectrum, and placed them in close relationship to inflammatory macrophages from the ascites of human ovarian cancer. The identified ercDC-specific gene expression profile was predictive for patient survival and suggests potential targets for therapeutic intervention that may help improve clinical efficacy of immunotherapy.

## 2. Materials and Methods

### 2.1. Tissues, Cells and Cell Culture

Tissue and blood collection were approved by the local ethics commission of the LMU München, and patients/donors consented to the donation. Tissue samples of histologically diagnosed clear cell renal cell carcinoma (ccRCC) (*n* = 15) were obtained from untreated patients who underwent surgery at the Urology clinic Dr. Castringius Planegg (Munich, Germany). Patient characteristics including TNM classification are shown in Appendix A. Fresh postoperative material was used to prepare cell suspensions and cryosections [27]. Peripheral blood mononuclear cells (PBMCs) from healthy donors (HD) were used to isolate monocytes (using CD14^+^ microbeads, Miltenyi Biotec, Bergisch Gladbach, Germany) and to sort CD1c^+^ DC and slanDCs using FACS (Aria IIIu from BD Biosciences, Franklin Lakes, NJ, USA). Before cell sorting, PBMCs were depleted of B and NK cells via CD19^+^ and CD56^+^ microbeads according to the manufacturer’s protocol (Miltenyi Biotec). M1- and M2-macrophages were generated from blood monocytes in vitro, as described [32]. Briefly, monocytes were cultured in 6-well plates (Nunc by Thermo Fisher Scientific, Waltham, MA, USA) in serum-free medium (5 × 10^6^/4 mL of AIM-V) supplemented with M-CSF (50 ng/mL; R&D Systems, Minneapolis, MN, USA) over 7 days followed by 18 h treatment with IFN-γ (20 ng/mL; R&D Systems) plus LPS (100 ng/mL; Sigma-Aldrich, Darmstadt, Germany) for M1-polarization, or IL-4 (20 ng/mL; PromoKine by PromoCell GmbH, Heidelberg, Germany) for M2-polarization.

### 2.2. Cell Sorting

ErcDCs and macrophages were sorted from ccRCC tissue-cell-suspensions stained with CD45-PeCy7, CD11c-APC, CD3-PB, CD209-PE (all BD Biosciences), CD14-PerCPCy5.5 (eBioscience, San Diego, CA, USA) and LIVE/DEAD^®^ Fixable Near-IR Dead Cell Stain Kit (Thermo Fisher Scientific). Sorting gates were set on CD209^+^CD14^+^ cells (ercDCs) and CD209^−^CD14^+^ cells (macrophages), among pre-gated CD45^+^, live, single CD11c^+^ CD3^−^ cells. CD1c^+^ DC and slanDCs were sorted from B- and NK-depleted PBMCs of healthy donors (HD) using anti-CD11c-PE, anti-CD3-PB (all BD Biosciences), anti-CD56-APC (Beckman Coulter, Brea, CA, USA), anti-CD19-PB (Dako by Agilent, Santa Clara, CA, USA), anti-CD1c-PeCy7 (Biolegend, San Diego, CA, USA), anti-slan-FITC (Miltenyi Biotec) and LIVE/DEAD^®^ Fixable Near-IR Dead Cell Stain Kit. The gating strategy and instrument parameters are in Appendix A. Gates were set very strictly, not covering the whole population, to avoid contamination with other cell populations. Cell population purity varied between 98–100%. Cells were directly sorted into 250 µL of RLT lysis buffer with ß-mercaptoethanol (RNeasy Micro Kit by Qiagen, Venlo, The Netherlands) using FACSAria IIIu (BD Biosciences), then homogenized (QIAshredder by Qiagen) and stored at −80 °C. Details about sorted cell types and numbers of biological replicates are listed in Appendix A**.**

### 2.3. Polychromatic Flow Cytometry

Polychromatic Flow Cytometry involved 1–5 × 10^5^ cells being incubated with respective antibodies for 30 min/4 °C and LIVE/DEAD^®^ Fixable Near-IR or Blue fluorescent Dead Cell Stain Kit, washed, optionally incubated with secondary antibodies and acquired at LSRII (BD Biosciences). In tissue-cell-suspensions, CD45 was used to identify leukocytes after exclusion of dead cells and doublets. Myeloid cells were selected based on CD11c (pan-myeloid marker in human) with exclusion of CD3^+^ T cells. Within CD11c^+^ cells, ercDCs and macrophages were distinguished as CD209^+^CD14^+^ double-positive cells (ercDCs) and CD14 single-positive cells (macrophages). For M1/M2-macrophages, live cells where selected and doublets were excluded before marker analysis. Antibodies are listed in Appendix A.

### 2.4. Immunofluorescence Histology and Confocal Microscopy

Cryosections were fixed with 4% paraformaldehyde (PFA, Merck, Darmstadt, Germany) and stained with primary and fluorescent-labeled secondary antibodies as described [27]. The antibody combinations were goat-anti-human VSIG4 (IgG by R&D Systems) and mouse-anti-human CD209/DC-SIGN (mouse IgG2a by SantaCruz Biotechnology, Santa Cruz, CA, USA) followed by secondary antibodies donkey anti-goat-A488 (Thermo Fisher Scientific) and rat anti-mouse-IgG-Cy5 Slides were mounted with ProLong^®^ Gold Antifade (Thermo Fisher Scientific). Fluorescence images were captured with a laser scanning microscope TCS SP5 (Leica Microsystems, Wetzlar, Germany) with settings as described [27].

### 2.5. RNA Isolation, Microarray Hybridization

RNA was prepared using RNeasy Micro Kit (Qiagen). RNA quantity and quality was assessed using a Nanodrop 1000 Spectrometer (Peqlab, Erlangen, Germany) and Agilent 2100 Bioanalyzer. From in vitro-generated cells (M1-, M2-macrophages) and CD14^+^ monocytes, three replica pools (each consisting of RNA from 5 different HD) were generated. RNA from flow-sorted cells was not pooled (Appendix A). RNA (30 ng of each replica pool; 0.5–15 ng of sorted cells) was subsequently amplified and converted into cDNA by a linear amplification method using WT-Ovation PicoSL System in combination with the Encore^®^ Biotin Module (both from NuGen, San Carlos, CA, USA). cDNA was hybridized to Affymetrix GeneChip^®^ Human Gene 1.0 ST Arrays.

### 2.6. Microarray Data Preprocessing and Probe Set Filtering

Raw intensity data were processed with R/Bioconductor (Bioconductor.org (accessed on 7 September 2022)). If not stated otherwise, functions were called with default parameters. We calculated normalized expression values for each study group (in-house generated and external data sets, see Appendix A) independently using Robust Multichip Average (RMA, library “oligo”) preprocessing including background correction and quantile normalization. Technical control probe sets as well as probe sets whose values did not vary between arrays (variance = 0) were excluded from all further analyses. Many-probe-sets-to-one-gene relationships were resolved by keeping only one probe set with the highest variance for each gene. Further analyses included only informative genes, which we defined as the group with the 50% most variable expression within the respective study group.

### 2.7. Combining Microarray Studies

For comparing transcript levels across studies, gene expression values were merged based on annotated gene entrez ids and study batch effects were removed using the COMBAT method [33] (library “inSilicoMerging”). We included all samples of a study in the merge and selected samples of interest afterwards.

### 2.8. Hierarchical Clustering and Heatmaps

Gene and samplewise hierarchical clustering of expression profiles used Euclidean distances and the complete agglomeration method. Heatmaps represent color-coded genewise standardized expression levels (mean = 0, standard deviation = 1; z-score).

### 2.9. Marker Genes

Nearest shrunken centroid classifiers [34] were constructed with the function “pamr.train” and cross-validated with the function “pamr.cv” (library “pamr”) (Appendix A) on informative genes of compared data sets. The classification threshold (1.88) was set such that the false positive classification rate was smaller than 20% and a preferably small number of genes was obtained.

GeneMANIA network analysis [35] was conducted based on the ercDC_ccRCC marker genes with default parameters (data as of May 2014).

Differentially expressed genes (DEGs) between ercDC_ccRCC&infMΦ_ascOvCa and the control group (all other samples/groups listed in Appendix A) were identified by a linear model using empirical Bayes moderated *t*-tests (R package “limma”) and Benjamini-Hochberg correction for multiple testing. DEGs were defined by an adjusted *p*-value < 0.05.

For Gene ontology enrichment (GO) analysis, hypergeometric *p*-values for enrichment or depletion of differentially expressed genes in GO categories of the group “biological process” were calculated with the function “hyperGTest”, Bioconductor package “GOstats”. Informative genes served as background and *p*-value threshold was set to 0.001.

Enrichment of differentially expressed genes within InnateDB [36,37,38] signaling pathway gene sets (from “KEGG”, “BioCarta”, “Reactome”, “NetPath”, “INOH” and “PID”) was defined with default settings at InnateDB.com (data as of May 2014). Informative genes served as background.

GSEA analysis was performed with GSEA 1.0 R-script 2014 of the Broad Institute with default parameters [39] (http://www.broad.mit.edu/gsea/ (accessed on 8 April 2014)).

For describing gene-expression-module-to-cell-type relationships, we first calculated median expression values per gene for each cell type. Eigengene values for gene expression modules as described [40] were calculated with the function “moduleEigengens” from the R-package “WGCNA” [41]. Eigengenes were correlated to cell types using Pearson’s method.

### 2.10. ercDC Ccore in the Cancer Genome Atlas (TCGA) and the Validation (Rostock) Cohort

Transcriptome profiling data (“HTSeq-FPKM-UQ”) of the TCGA ccRCC cohorts (KIRC, LAML) were downloaded from the Genomic Data Commons Portal https://gdc-portal.nci.nih.gov/ (accessed on 9 December 2016 and 15 December 2016, respectively). Clinical data were obtained from the same platform on 10 October 2016 and 3 November 2016, respectively. To be used in this study, TCGA samples had to meet the following criteria: Patients with neoadjuvant therapies (“history_of_neoadjuvant_treatment”) were excluded. Moreover, only subjects with available survival data were considered (overall survival, OS, for the LAML cohort; cancer-specific survival (CSS) as defined in [42] for the ccRCC cohort). Follow-up time was required to be greater than 0. Missclassified patients [42,43] revealed by cluster analysis and/or by re-evaluation of tissue histology were also discarded from the TCGA ccRCC cohort. Characteristics of the final TCGA ccRCC cohort (*n* = 442 patients) are summarized in Appendix A.

The validation cohort included a collection of selected G1 and G3 ccRCC tissues (*n* = 14 each) and corresponding normal kidney tissue (*n* = 14) microarray data (“Rostock cohort”, Array GeneChip HG U133 Plus 2.0, Affymetrix) [44,45]. Clinical data and follow up are found in Appendix A.

An ercDC score was established by gene expression deconvolution. Expression values of the 61 marker genes in the ercDC arrays were first collapsed by taking the median. FPKM-UQ expression values in the TCGA ccRCC cohort were log-transformed (log2(x + 1)). Subsequently, using the marker gene set, a simple linear regression model was fit with “expression in ercDC cell type” as predictor and “expression in sample” as regressor. Finally, the slope of the linear model constituted the ercDC score of a sample. Conditional inference trees from R-package partykit_1.1-1 [46,47] were used to identify groups with significantly varying CSS curves in both the TCGA ccRCC cohort and the validation cohort. The *p*-value criterion of the conditional inference tree method was weakened (0.1) and the minimum group size was set to 10. Kaplan–Meier curves and corresponding log-rank tests using R-package survival_2.40-1 [48] were applied for survival analysis. Further, R-package coin_1.1-3 [49] was used to perform a linear trend test between ercDC score and tumor grade.

### 2.11. Public Access to Raw Data of Data Sets Analyzed in this Paper

Our data sets of human ercDCs and macrophages from ccRCC tissue, myeloid cells from PBMCs and in vitro-generated M1- and M2-macrophages are accessible via super series GSE108312.

## 3. Results

### 3.1. The ercDC Transcriptional Profile Identifies Them as a Unique Myeloid Subset within the Macrophage Spectrum

The CD14^+^CD209^+^ ercDCs and CD209^−^CD14^+^ macrophages were sorted from ccRCC tissue cell suspensions (Appendix A) using flow cytometry and subjected to genome-wide gene expression analysis. Reference transcriptomes were generated from sorted blood monocytes (CD14^+^), slanDC and CD1c^+^ DC (all from PBMCs of healthy donors (HD)), and from in vitro-polarized M1- and M2-macrophages as described [11,32] (Appendix A). CD1c^+^ DCs and slanDCs have been described to exhibit proinflammatory DC characteristics with IL-12 production and T cell priming capacity [50,51]. They were used as DC reference cell types in the comparison, to substitute for interstitial DCs, i.e., CD209^+^CD14^−^ cells, which could not be sorted from ccRCC tissue cell suspensions due to low cell frequency. In comparison to ercDCs, they were expected to be opposite to the ercDCs, for which a lack of IL-12 secretion and priming capacity has been observed [27].

Previously, using flow cytometry analysis, it was difficult to assign ercDCs to either macrophages or DC subgroups as they co-expressed markers of both cell types, i.e., CD209, CD14, and the co-stimulatory molecules CD80, CD86 and CD40. Analysis of their transcriptome with respect to core macrophage and DC genes [40,52,53] now revealed that ercDC_ccRCC strongly expressed most of the human macrophage-associated core genes and showed expression of some of the established DC-associated core genes (Figure 1A). Protein expression of selected macrophage (CD64A, CD14, MerTK, CD32A) and DC markers (ANPEP/CD13, FLT3) was confirmed by flow cytometry (Figure 1B,C).

The assumption that ercDCs more closely represent a macrophage rather than a DC subtype was confirmed by analyzing transcription and growth factors from a second MPS classification scheme [54]. Macrophage-associated factors MAF, MAFB, CREG1 and CSF1R were robustly expressed, DC-associated transcription factors IKZF1, BCL6 and IRF4 showed weak expression (Appendix A).

It is recognized that macrophages represent a continuum of different subtypes, wherein M1- (classical) and M2- (alternatively activated) macrophages are positioned at the opposing ends of the polarization spectrum [12,55]. In addition, tissue macrophages are extremely heterogeneous and may adopt specialized functions as they respond to a variety of signals that change during homeostasis and inflammation [5,15,16]. A classification system based on the fundamental homeostatic macrophage activities—host defense, wound healing/tissue modulation and immunoregulation—has been established to help address this complexity [10]. To position the ercDC_ccRCC within these classification schemes, we analyzed the transcriptome with regards to gene lists associated with biologic function [10,56,57] (Appendix A) including also an invasive signature gene list [57] which we supplemented with key angiogenic genes taken from the GSEA MSigDB database (Appendix A). Notable was the strong expression of the prototypic M1-gene CD64A [7,58] (Figure 2A) and M2-associated markers, including MerTK, CD204, CD206 and CD36 (Figure 2B). Validation of surface expression was performed for CD64A (Figure 1B and Figure 2C) and M2-markers CD204, CD206 and CD36 (Figure 2B, bottom). Single cell flow cytometry of ccRCC tissue suspensions confirmed on protein level that the majority of CD14^+^CD209^+^ cells co-expressed CD64A (M1-marker) with MerTK and MSR1/CD204 (both M2-markers) (Figure 2C) indicating that ercDCs are mosaic cells on the single cell level.

ErcDC_ccRCC were found to express a number of chemokines (CXCL9, CXCL10 and CXCL11) that are characteristic of M1-macrophage subtypes associated with host defense activity and help in the recruitment of Th1-polarized immune cells (Figure 2A). ErcDC_ccRCC also expressed genes linked to wound healing and tissue remodeling (i.e., STAB1, FN1, F13A) (Figure 2D), thus resembling CD11c^LO^ MΦ_decidua, which have been linked to wound healing and tissue remodeling [8,31,59,60]. In addition, ercDCs showed a strong signature of angiogenesis and invasion-associated genes (Figure 2E, top). These included genes associated with the recruitment of proinflammatory monocytes, e.g., CCL2, CCL8 and NRP1, as well as proinflammatory factors such as TNF, IL6 or IL1B. Genes involved in degradation of the extracellular matrix (e.g., MMP2, MMP9), hypoxia regulated genes (HIF1A) and proangiogenic genes (GPNMB, VEGFA, IGF1) were also part of the ercDC profile. High mRNA expression of MMP2 and MMP9 confirmed our previously described protein data and, together with the angiogenic signature, supports the hypothesis that ercDCs help promote tumor growth [27]. Of note was the robust expression of VSIG4 in ercDC_ccRCC. Surface protein expression of VSIG4, GPNMB, NRP1 and CD9 was validated by flow cytometry (Figure 2E, bottom).

The ercDC transcriptome also contained a number of immunoregulatory genes (i.e., MAF, VSIG4, TREM2, CD206) (Figure 2F) [9,61,62]. The strong expression of factors linked to the induction of T cell tolerance, MAF [63] and VSIG4 [64], support an immunoregulatory role for ercDCs in ccRCC tissue. VSIG4 was strongly expressed on the protein level. Multiparameter immunofluorescence histology of ccRCC tissues showed that the majority (85–98%) of CD209^+^ cells co-expressed VSIG4. Tumors of advanced stage harbored more CD209^+^VSIG4^+^ cells (abs. median number 45.7, range 21–75) than those of earlier stages (median number 35, range 31–58) (representative images, Figure 2G). Well described markers of immunoinhibition (PD-L1/B7-H1 and PD-L2/B7-DC) [65,66] were only marginally expressed. TIM-3 showed weak expression, B7-H3 was strongly expressed (Figure 2H).

### 3.2. ErcDCs Have a Gene Expression Signature Similar to Inflammatory Macrophages from Ascites of Ovarian Cancer with Characteristics of Immune Tolerance

Renal tubulo–interstitial DCs are described to act as sentinels maintaining homeostasis and protecting the renal tubuli from immune-induced injury through tolerogenic mechanisms [27,67]. To investigate whether ercDCs arising in the tubulointerstitial milieu of ccRCC similarly exhibit tolerizing features that might confer tumor immune protection, we conducted a global analysis across published transcriptomes. The reference data comprised human myeloid cell types from blood and various non-lymphoid tissues, including myeloid subsets originating from tissues with described tolerogenic milieus (detailed information together with the rationale for their selecting are in Appendix A). In brief, the subsets tested were CD11c^HI^ and CD11c^LO^ decidual macrophages [31], three DC subtypes from the lamina propria distinguished by their expression of CD103 and Sirpα (CD103^+^Sirpα^+^ DCs, CD103^−^Sirpα^+^ DCs, CD103^+^Sirpα^−^ DCs) [68], alveolar macrophages from non-smokers, smokers and COPD or asthma patients [69,70], human tumor associated macrophages (TAMs) from gastrointestinal stromal tumors (GIST) [71], two myeloid cell types from the ascites of ovarian cancer patients (inflammatory macrophages (infMΦ_ascOvCa) and inflammatory DCs (infDC_ascOvCa)) [72,73], and CD141^+^ DCs from peripheral blood [74] representing DCs with the capability of cross-presentation and activation of CD8^+^ T cells [75,76]. We supplemented the published data sets with in-house expression profiles of PBMC-derived monocytes, slanDCs and CD1c^+^ DCs.

Hierarchical clustering revealed similarity of ercDC_ccRCC with MΦ_ccRCC, infMΦ_ascOvCa and infDC_ascOvCa (Figure 3A). CD11c^LO^ MΦ_decidua, TAM_GIST, CD103^+^Sirpα^+^DC_gut and MΦ_asthma_avlung formed a separate subgroup. Blood-derived cell types, together with CD103^+^Sirpα^−^ DC_gut and CD103^−^Sirpα^+^ DC_gut clustered distinct from all other cell types. CD141^+^ DC_blood more closely resembled CD103^+^Sirpα^−^DC_gut, confirming the similarity in characteristics previously described, including expression of markers associated with cross-presentation [68]. Principle component analysis (PCA) (Figure 3B) provided further evidence that ercDC_ccRCC are most similar to MΦ_ccRCC and infMΦ_ascOvCa, and are clearly different from blood-derived cells.

To assess if expression states of specific genes can distinguish ercDC_ccRCC from other myeloid subtypes, we used the nearest shrunken centroids method (NSCM) [34], a supervised machine learning approach suited to define subsets of genes that best characterize specific cellular states. Feature selection on the classifier that we trained to predict ercDC_ccRCC revealed 61 marker genes as predictive for ercDCs (Appendix A). Hierarchical clustering of all myeloid cell types based on expression of the 61 marker genes (Figure 4A) showed that ercDC_ccRCC clearly separated from the other human cell types analyzed with the exceptions of infMΦ_ascOvCa and CD11c^LO^ MΦ_decidua. Despite originating from the same tissue, MΦ_ccRCC were positioned in a different cluster together with infDC_ascOvCa, and were thus not classified as ercDC_ccRCC. The tumor-associated macrophages from GIST (TAM_GIST), which are described as an antitumoral M1-like TAM subtype, did not show similarities with the ercDC_ccRCC profile. Blood-derived myeloid cells, slanDC_blood, CD1c^+^DC_blood and Mono_blood, exhibited the strongest differences to the ercDC_ccRCC profile, with an almost inverse expression of many of the marker genes.

Of the 61 marker genes, 39 were upregulated while 22 were downregulated in ercDC_ccRCC as compared to the other cell types (Figure 4A, Appendix A). As expected, CD209/DC-SIGN, which was used to distinguish ercDC_ccRCC from MΦ_ccRCC in flow cytometry, was present in the marker gene list and showed increased expression (Figure 4A). The macrophage core gene *SEPP1*, the M2-associated gene *MAF* and the M1-associated gene *CXCL9* were present in the ercDC_ccRCC marker gene list, as well as genes associated with immunoinhibitory and proangiogenic functions, like *GPNMB* and *NRP1*.

Given the strong similarity of ercDCs and infMΦ_ascOvCa, we defined a list of DEGs between these two cell types and the other myeloid cell types. The genes with significantly different expression (FDR < 0.05) in ercDC&infMΦ_ascOvCa as compared to the other samples (control group) are shown in Figure 4B and were designated ercDC_ccRCC DEGs. They include 788 genes, with 431 showing upregulation and 357 downregulation (Appendix A). The DEG list includes 54 of the 61 marker genes (89%), which are shown in bold letters with yellow background in the volcano plot (Figure 4B). Most of the marker genes showing the strongest and most significant expression differences were upregulated DEGs, e.g., *CD209/DC-SIGN*, *FOLR2*, *GPNMB* and *SEPP1*. The upregulated DEGs included genes associated with anti-inflammatory function and macrophage recruitment. These include *CD204/MSR1*, *MERTK*, *CD163*, *CCL2*, *CCL8* and *CCL18* (Figure 4B). Downregulated DEGs included ercDC_ccRCC marker genes (*FAM65B*, *FGR, CFP*) and DC-associated genes, including *BCL11A* and *FLT3*.

Superposition of M1- and M2-associated genes (Appendix A) on the list of informative ercDC_ccRCC genes again illustrated the mosaic expression of M2- and M1-associated genes by the ercDC_ccRCC myeloid subtype (Figure 4C). Among the most significantly upregulated genes were the M2-associated genes *CD209*/*DC-SIGN, SLCO2B1, SLC38A6, SEPP1, CCL18, MAF*, *MS4A4A* and *IGF1*, but also the M1-associated genes *IL2RA* and *CXCL9*. These M1/M2-associated genes also belonged to the ercDC_ccRCC marker genes. Overall, 2.6% of the M1-associated genes and 12.9% of the M2-associated genes were among the ercDC_ccRCC marker genes. Moreover, 23.7% of M1-associated genes and 45.2% of M2-associated genes were part of the ercDC_ccRCC DEGs. GSEA analysis provided evidence of significance for the difference in expression of the M2-gene set (*p* = 0.02) and enrichment of the M1-gene set (*p* = 0.09) in the ercDC_ccRCC&infMΦ_ascOvCa. This suggests that ercDC_ccRCC represent a hybrid myeloid subtype with mosaic features of M2- and M1-polarization.

To identify enriched biological processes associated with the DEGs, GO term analysis was employed. The results showed “response to wounding” and “inflammatory response” as the most significantly scored categories (Figure 5A). These effector processes correspond with the described inflammatory milieu in RCC [77,78] and the well-recognized role for renal macrophages in normal tissue homeostasis and wound healing [79,80]. The third most enriched category was “defense response”, underscoring a bactericidal activity of ercDC_ccRCC, which agrees with the general expression of M1-associated genes.

The results of InnateDB and GSEA (based on the data bases “KEGG”, Reactome” and “Biocarta”) analyses overlapped with GO term analysis (Appendix A). Most of the enriched pathways identified are associated with the complement system, lipid metabolism and modulation of the extracellular matrix. Deregulated lipid metabolism has been described to promote the development and progression of RCC [81].

GeneMANIA network analysis was conducted to better characterize the functional network of ercDC_ccRCC marker genes, and to identify functionally related genes (Figure 5B). The identified related genes (grey) included many macrophage-associated genes, particularly those of M2-macrophages and their immunoinhibitory function, *CD163*, *CD206*/*MRC1*, *CD14* and *VSIG4* (yellow circles in Figure 5B).

### 3.3. ErcDCs Are Distinct from Blood-Derived Monocytes from RCC Patients

Inflammatory blood monocytes can act as precursors for TAMs [82,83,84]. Chittezhath et al. reported that blood-derived monocytes from RCC patients display a tumor-promoting transcription profile [85]. Hierarchical clustering and PCA evaluation including the RCC-monocyte transcriptome (designated as Mono_RCC_blood) clearly separated the Mono_RCC_blood from the ercDC_ccRCC and clustered the Mono_RCC_blood with CD11c^HI^ MΦ_decidua and CD103^−^Sirpα^+^ DC_gut (Figure 6A,B).

Tissue-specific gene expression can obscure cell type-specific profiles. However, even after exclusion of blood-specific genes (“tissue preferential expressed gene list (blood genes)” [86]) Mono_RCC_blood remained clearly distinct from ercDC_ccRCC and retained their similarity with CD11c^HI^_MΦ_decidua and CD103^−^Sirpα^+^ DC_gut (Appendix A). Clustering based on the ercDC_ccRCC marker genes positioned the Mono_RCC_blood distant to ercDC_ccRCC in a subcluster together with CD11c^HI^ MΦ_decidua (Figure 6C), similar to the hierarchical cluster analysis on all informative genes (Figure 6A). Identical clustering was observed after the exclusion of blood-specific genes (Appendix A).

Mono_RCC_blood resembled ercDC_ccRCC in their expression of M1-associated genes, but were more similar to CD1c^+^_DC_blood in their expression of M2-associated genes, while ercDC_ccRCC clustered with M2-macrophages (Figure 6D). Removal of the blood-specific genes again did not change this profile (Appendix A). Chittezhath et al. reported that Mono_RCC_blood derived protumoral activity through an interleukin-1 receptor (IL-1R)-dependent mechanism. This pathway was, however, less expressed in ercDC_ccRCC compared to Mono_RCC_blood (GSEA, *p* = 0.0001) (Appendix A).

### 3.4. The ercDC_ccRCC Polarization Profile Reveales Distinct Tissue Imprints

It is acknowledged that the phenotype and function of tissue-resident macrophages is robustly influenced by factors present in the tissue micromilieus [13,14,15,16]. To investigate the potential influence of the ccRCC milieu on the gene expression profile of ercDC_ccRCC, predefined stimulus-specific human gene sets were used that are known to be induced by distinct macrophage activation signals [40]. We correlated the module eigengenes (ME) of the stimulus-specific gene sets (so-called modules) with the gene expression profile of ercDC_ccRCC, various human myeloid cell types from non-lymphoid tissues, including Mono_RCC_blood and in vitro-generated M1- and M2-macrophages (Figure 7, Appendix A). The ercDC_ccRCC clustered closest with infMΦ_ascOvCa, confirming the strong relationship seen previously using the NSCM. Macrophages from the ccRCC tissues (MΦ_ccRCC), M2-MΦ_in vitro and infDC_ascOvCA clustered within the same group (no. 4). All cell types from this group also showed largely negative correlations with modules in cluster A, linked to signals associated with the cytokine IL-4. On the other hand, they showed predominantly strong positive associations with modules in cluster B. The modules of cluster B were generally linked to signals from glucocorticoids (GC), palmitic acid (PA), prostaglandin E2 (PGE_2_) or a combination of TNF, PGE_2_ and P3C (Pam3CysSerLys4, TLR2-ligand, TPP) (in orange letters). Other modules in cluster B are linked to M1-polarizing stimuli (IFN-γ, TNF) (in brown letters); accordingly, a very strong positive correlation was found with in vitro-polarized M1-macrophages (M1_MΦ_in vitro) and a negative correlation with M2_MΦ_in vitro. ErcDC_ccRCC also correlated negatively or showed no correlation, while infMΦ_ascOvCA, infDC_ascOvCA and MΦ_ccRCC showed some positive correlation. Overall, Mono_RCC_blood showed much weaker and different polarization than ercDC_ccRCC, likely reflecting their exposure to different local milieus.

In summary, the data provide evidence that the characteristic transcriptional profile of ercDCs may be induced in part by GC, PGE_2_, PA and TPP, and low abundance of IL-4. This is consistent with described characteristics of the RCC tissue milieu, featuring GC [87], PGE_2_ [88] and TNF [89,90] and marginal IL-4 [91]. PA has been described as playing a role in kidney fibrosis [92]. The emerging imprinting stimuli are also consistent with the marker profile determined using NSCM and the upregulated DEGs. PGE_2,_ for example, is described as inducing the transcription of *MSR1*/*CD204* [93], a gene also strongly expressed in ercDC_ccRCC. Moreover, PGE_2_ and GC are known inducers of *CD163* [94,95], another marker strongly upregulated in ercDC_ccRCC. In addition, GC regulates the expression of *MERTK* as well as *C1QB*, *CCL8*, *VSIG4* and *FCN1*, all of which belong to the ercDC_ccRCC marker genes or DEGs.

### 3.5. Expression of ercDC Marker Genes in ccRCC Tissue Is Predictive of Patient Survival

We have previously shown that CD209^+^ cell numbers were higher in advanced ccRCC tumors with poor prognostic tumor stage [27]. Here, using an ercDC score based on the marker gene expression, we support this finding in two independent cohorts, the Cancer Genome Atlas (TCGA) cohort of 442 ccRCC samples (Appendix A) and the Rostock cohort, which is a preselected arrangement of 28 primary ccRCC tumor tissues equal representation of G1 and G3 histology grades (Appendix A). The ercDC score was higher in ccRCC tissue compared to control non-tumor renal tissues and it enriched with increasing tumor grade (Figure 8A,C). Significantly, in both cohorts, patients with high ercDC scores showed decreased cancer-specific survival (CSS) compared to patients with low ercDC score (TCGA: HR = 1.8, *p* = 3.0 × 10^−2^; Rostock: HR = 4.8, *p* = 8.8 × 10^−3^).

Early stage tumors (G1), as compared to later stage tumors (G3), not only showed a lower ercDC score, but by trend also had a lower cytotoxic infiltrate, approximated by the mean expression levels of CD8A and NKG7 (natural killer cell granule protein 7 expressed by NK cells and CD8 T cells in ccRCC [22]) (Figure 8E,G).

Correlation analysis of ercDC score and cytotoxic infiltrate highlights a parallel expansion of cytotoxic infiltrate and ercDC score (Pearson correlation coefficient = 0.42). This relationship, however, was not proportional as the loess curve fitted using the standardized values was above the diagonal at lower ercDC values and below the diagonal at higher ercDC values (Figure 8I). Thus, the cytotoxic infiltrate dominated the ercDC score in earlier stage tumors while the ercDC score offset the cytotoxic infiltrate in the later stage tumors, as seen also by the calculated ratios (Figure 8F,H). This relationship may explain, in part, the finding that a larger CD8 T cell infiltrate is not beneficially associated with survival in ccRCC, contrasting most other tumors [96].

The relationship is consistent with our previous histology work where a proportionally lower CD8 cell count was found in tumors with high ercDC numbers [27]. While other reasons may cause this relationship, ercDCs may actively contribute by curbing the cytotoxic immune cell recruitment through reducing Th1 recruiting chemokines previously observed in in vitro T cell/tumor cell co-cultures.

The more favorable composition of the infiltrate in G1 tumors might contribute to the prognostically better outcome of patients with early stage tumors. In G1, fewer ercDCs may be sufficient for immunoinhibition in the context of a lower cytotoxic infiltrate and ensuing interaction between T cells, macrophages and tumor cells, which can lead to tumor progression through macrophage-secreted tumor promoting factors, such as CCL8, CCL18 and MMP9, which are part of the ercDC signature.

## 4. Discussion

Since the 1970s, RCC has been recognized as an immune-responsive tumor with a well-documented sensitivity to T cell attack [97,98,99]. Yet, RCC patients do not benefit substantially more from the T cell activating immune checkpoint blockade therapy compared to patients with tumors previously considered to be non-immunogenic [23,25,26,100,101]. This suggests the existence of additional layers of tumor-mediated immunosuppression beyond targeted T cell checkpoints that hamper antitumor immune responses. These layers need to be addressed to improve the efficacy of cancer immunotherapy.

Here we describe a myeloid cell type that was strictly resident to non-lymphoid tissue and preferentially found in the tumor center of ccRCC. Its transcriptome exhibited a unique mosaic expression pattern encompassing gene transcripts of various macrophage polarization states. Thereby our analysis confirms and extends previous studies that have reported diametrically polarized macrophages in RCC tissue [102,103,104,105]. The expressed genes provide evidence for tumor-promoting qualities as well as immunoinhibition that may translate to T cell dysfunction. A high ercDC score in ccRCC tissue was found to be strongly associated with poor patient survival. Targeting ercDCs particularly in situations where tumors have high ercDC content may expand the range of patients that can be effectively treated with immunotherapy.

The ercDC transcriptome revealed details regarding their functional polarization and positioning within the MPS. Contrary to our previous assignment of the ercDCs to the DC lineage, the transcriptome suggests that ercDCs belong to the macrophage lineage. Notably, they showed combined features of M1-macrophages and M2-macrophages as well as gene signatures associated with wound healing and tissue remodeling, immunoregulation and bactericidal effector activities. The ercDC attributes could be clearly related to the inflammatory milieu of RCC tissue [77,78] and the general role of macrophages in tissue homeostasis and wound healing in kidneys [79,80]. Observed imprints of RCC characteristics include altered lipid metabolism [106,107] and the accumulation of immune complexes with associated complement activation, which is also seen in many inflammatory renal diseases [108,109]. A relationship between the ercDC from RCC tumor tissue and myeloid cells from chronic inflammatory kidney pathologies was previously suggested based on the triple marker staining of CD14, CD209 and CD163 [27,110]. CD14^+^CD209^+^CD163^+^ triple positive cells are also described in the human dermis, in a murine leprosy model, and in the decidua of early human pregnancy [27,111,112,113]. The presence of these cells was linked to deviated immune responses, allowing bacterial or embryo persistence. As this seemed to parallel the situation of tumors, available transcript data were compared to the ercDC transcriptome. However, despite the conceptual similarity, the myeloid cells from these tissues did not appear to be closely related to our ercDCs. Yet, the recently available single cell sequencing data describing the immune landscape of lupus nephritis revealed five clusters of myeloid cells [114], whereby cells in cluster 4 (CM4) were found to express the index markers of ercDCs, CD14, CD209 and CD163, together with further ercDC marker genes *CD64, VSIG4, MSR1, STAB1, MerTK, FOLR2* and *C1Q*. The transcript profile of CM4 cells did not match any cell type found in peripheral blood, suggesting kidney residency of this myeloid cell type. Likewise, we were not able to detect CD209^+^CD14^+^CD163^+^ ercDCs among PBMC of healthy donors or patients by flow cytometry (data not shown) and the ercDC score was very low in samples of acute myeloid leukemia of the TCGA collection compared to ccRCCs (not shown). Moreover, by hierarchical clustering of the transcriptome data of ercDC_ccRCC and other myeloid cell types, ercDCs clearly separated from all blood-derived myeloid cells, including blood-derived monocytes from RCC patients, which were described to potentially represent precursors of RCC-TAMs [85]. The Mono_blood_RCC cell subset was clearly classified as blood-derived myeloid cells by hierarchical clustering with a transcriptome distinct from tissue-derived cell types, including ercDCs_ccRCC, even after elimination of blood-specific genes. Recent studies of breast, endometrial and lung cancer-derived myeloid cells confirm our notion that myeloid subsets in patient blood show limited overlap with those in their tumors [115,116].

A comparative analysis across published transcriptome databases of human myeloid cell types from blood and non-lymphoid tissues using the NSCM method identified an inflammatory macrophage subtype from the ascites of ovarian cancer (infMΦ_ascOvCa) as the closest relative to ercDC_ccRCC. Described features of ovarian TAMs include the expression of M2-markers (CD14, CD206, CD11b, CD204), select M1-markers (CD86, TNF) [73,117,118,119,120,121], and expression of the immunosuppressive and anti-inflammatory chemokine CCL18. These expression patterns are analogous to those seen in ercDC_ccRCC. Interestingly, cytokines shown to induce the ercDC phenotype in vitro, IL-6, CXCL8/IL-8 and VEGF [27], are also present in ovarian ascites [122,123,124]. Thus, the relatedness of ercDC_ccRCC and infMΦ_ascOvCa may be explained in part by their exposure to a related tissue milieu. In addition, PGE2, GC, TNF, PA and TLR2-ligands were identified as ercDC_ccRCC polarizing factors. PGE2 is an important factor in RCC biology [88]. The enrichment in PGE2 and GC is linked to the expression of CD163 [94,95] in ercDCs. In addition, GC has been described to induce MerTK [125] and VSIG4 [126], and may therefore be the reason for the expression of those markers in ercDCs. The observed close relationship of ercDCs to the infMΦ_ascOvCa may be explained in part as both are myeloid cells from epithelial tumors. The TAM_GIST are from a mesenchymal tumor, and the dissimilar transcriptional data might result from the different tissue milieu. It is, however, interesting that of the two myeloid cell types from the ovarian cancer (infMΦ_ascOvCa and infDC_ascOvCa), only the infMΦ_ascOvCa was computationally associated with ercDCs, showing stronger relationship than the MΦ_ccRCC. This suggests that within the tumor microenvironments, different niches exist which shape the transcriptional profile of the residing myeloid cells to become dissimilar to an extent that they more closely match to cells in a different tissue. Since our analysis was performed, additional transcriptomes of human myeloid cells from non-lymphoid tissue have become available, including those of human breast [115,127] and hepatocellular carcinomas [128]. ErcDC share marker expression with these macrophages, including FOLR2, CCL8, MMP2 and MMP9 as well as APOE and SEPP1. It will be exciting to learn if the closeness seen with the infMΦ_ascOvCa can be expanded to other macrophage populations as the field develops and we learn more about the biology at work. As an overarching conclusion of current publications, our data of ercDC transcriptome confirms the notion that the profile of tissue macrophages is very complex, integrating markers of various polarization states.

Due to our technical setup, we cannot formally exclude that this mosaic expression pattern arises from the assembly of multiple cell types with different functions. However, our analysis using polychromatic flow cytometry showed that the cells expressing the ercDC index markers, CD209 and CD14, co-expressed the M1-marker (CD64) and M2-markers (CD204 and MerTK). A separation of these markers over multiple cell populations was not evident. This supports the interpretation that the ercDCs are a unique cell type on the single cell level with mixed macrophage polarization. Chevrier et al. analyzed macrophages from RCC tissues by highly multiplex mass cytometry [129]. Unfortunately, the analysis did not include the CD209 marker nor the other relevant ercDC signature genes identified here, thus precluding a comparative analysis of our ercDCs to the 17 TAM phenotypes identified in this mass cytometry study. Mass cytometry is a powerful approach, but as a front-line analysis tool, it is disadvantaged by its bias towards previously characterized markers. Recently, single cell sequencing was applied to analyze the immune landscape of RCC [102,104,105]. The results support our finding that RCC tissues harbor macrophages with pro- and anti-inflammatory polarized phenotypes. Moreover, they support our observation that specifically polarized myeloid cells are associated with poor patient prognosis, and additionally provide evidence that RCC tissues with a high presence of macrophages are resistant to checkpoint blockade therapy. While there are certainly additional mechanisms related to tumor progression and therapy resistance, such as insufficient CD8 or NK cell counts or T cell dysfunction or inhibiting tissue factors, such as lactic acidosis [27,130,131,132], the finding by Braun et al. provides support for a role of macrophages as a mechanism of immune therapy resistance [102]. Observed contacts between macrophages and exhausted T cells suggest a T cell inhibitory communication that may be mediated by surface molecules. In this respect, our report of a novel series of proteins expressed on ercDCs is of interest. Especially, markers such as VSIG4 [64,133,134,135] and GPNMB [136,137], are discussed in the literature in the context of T cell inhibition and cell cycle arrest. While PD-L1, PD-L2 and TIM-3 were only marginally expressed by ercDCs, these new markers represent promising targets to moderate the communication that ercDC might mediate through T cell contact. NRP1 [138] and CSF-1R are additional possible targets expressed by ercDCs, for which therapeutic reagents are currently in clinical studies (https://clinicaltrials.gov/, accessed on 7 September 2022). The relevance of these markers for functional alteration of a T cell antitumor immune response will need to be assessed in experimental settings. Knockdown of these newly identified markers may reveal if ercDCs can be modulated to participate in productive antitumor immune response. Combined with checkpoint blockade or other therapeutic strategies, targeting these proteins may improve treatment outcome, especially for tumors with a high ercDC score.

## 5. Conclusions

Since the time of Metchnikoff, phagocyte biology has evolved tremendously. The diversity and plasticity are now acknowledged hallmarks of phagocytes and, functionally, they have taken center stage in tissue homeostasis and immune regulation. Our findings support the continuum of myeloid cell polarization and indicate the strong relationship between cell phenotype and tissue-derived signals. Single marker analysis is no longer adequate to assign phagocyte identity. Based on the expression of CD209, which is a marker of immature interstitial DCs in humans [28,139], and the association with murine kidney immune regulatory DCs [29,140], ercDCs were previously thought to represent a DC phenotype. The now described transcriptome identifies them as a macrophage phenotype with the mosaic presentation of markers related to diverse functional attributes ranging from antitumoral activities to tumor-promoting qualities as well as immunomodulation. Based on this now available information, we suggest changing the name to ercMP (*enriched-in-renal-cell-carcinoma* macrophages) for future reference. The high ercDC score in ccRCC tissue was found to be strongly associated with poor patient survival, showing prognostic value and suggesting that ercDCs may be targets for therapeutic intervention. Among the transcriptome, a novel series of potential targets was identified. As a correlation may not necessarily equate to causality, the potential role that ercDCs might play in the regulation of the antitumor response requires testing in experimental models. Targeting ercDCs in combination with immunotherapy, particularly in situations where tumors have high ercDC content, may expand the range of patients that can be effectively treated with immunotherapy.

## Figures and Tables

**Figure 1 cells-11-03289-f001:**
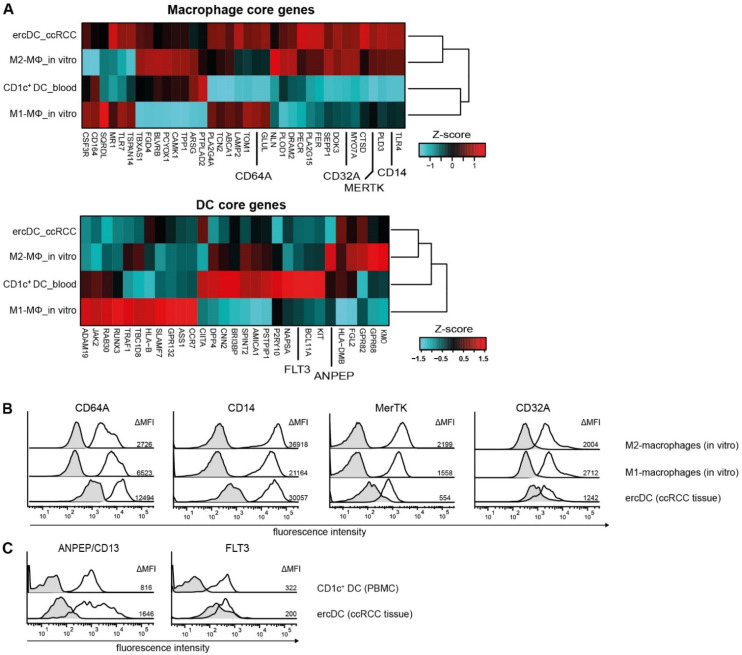
Human macrophage and DC core genes in ercDC_ccRCC. (**A**) Clustered heatmaps depict the relative gene expression levels of human macrophage and DC core genes comparing ercDC from ccRCC tissue (ercDC_ccRCC), in vitro-generated M1- and M2-macrophages (M1−ΜΦ_in vitro, M2−ΜΦ_in vitro), and CD1c^+^ DC from blood (CD1c^+^ DC_blood). Genes whose expression was validated on protein level (**B**,**C**) are set apart. (**B**,**C**) Protein surface expression by flow cytometry. In vitro-generated M1- and M2-macrophages, PBMCs and ccRCC tissue cell suspensions were stained with marker combinations and gated on ercDCs (CD209^+^CD14^+^ cells among CD45^+^ live single CD3^−^CD11c^+^ cells of ccRCC cell suspension), CD1c^+^ DCs (CD1c^+^ cells among live single CD3^−^CD11c^+^ of PBMCs) and on M1- and M2-macrophage population among live single cells. Depicted are representative histograms of human macrophage markers CD64A, CD14, MerTK, CD32A (**B**) as well as DC markers ANPEP/CD13 and FLT3 (**C**) from at least 6 different patient tissues or PBMCs. Black line histogram: specific staining, gray filled histogram: isotype or fluorescence minus one (FMO) control staining. Numbers indicate the control-corrected median fluorescence intensity (MFI).

**Figure 2 cells-11-03289-f002:**
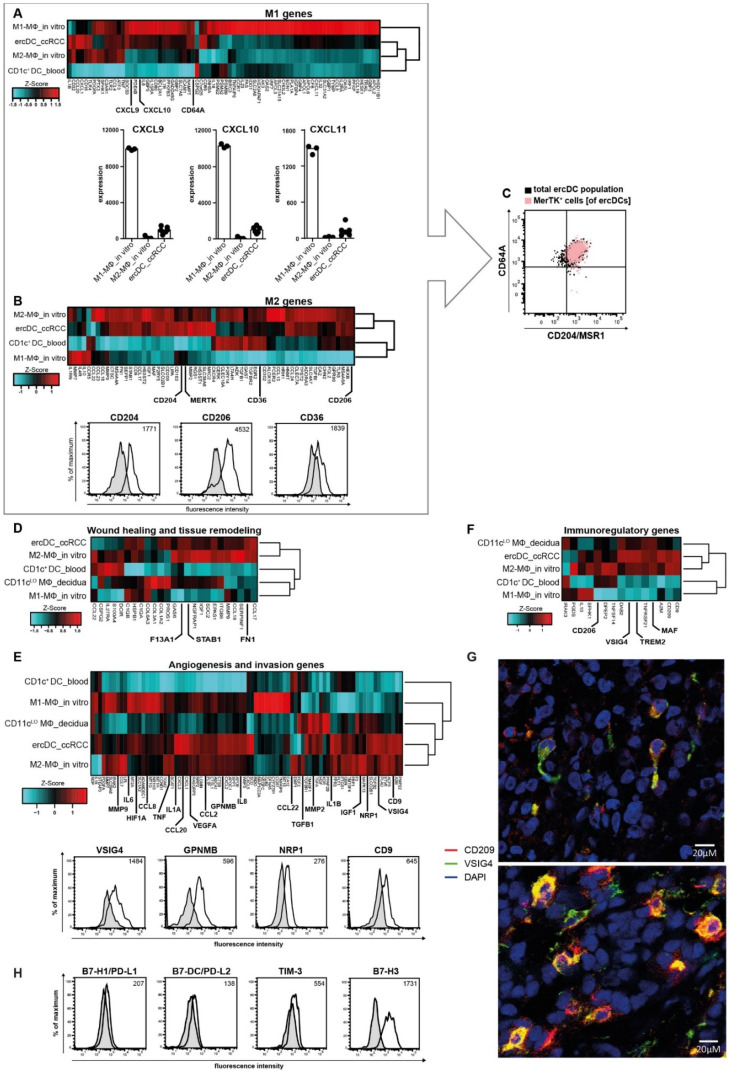
DC- and macrophage-associated genes and genes associated with angiogenesis and invasion in ercDC_ccRCC. (**A**,**B**) Clustered heatmaps depicts relative expression levels of genes associated with M1- and M2-macrophage polarization. See Appendix A for M1- and M2-genes. (**A**, bottom) Linear expression of chemokines associated with host defense activity of macrophages. M1-macrophages are the positive reference. Bars: median of each group; symbols correspond to individual array replicates of a cell type. (**B**, bottom) Surface protein levels of indicated markers validated by flow cytometry. (**C**) Dot plot of polychromatic flow cytometry demonstrates co-expression of M1- (CD64A) and M2-macrophage markers (CD204/MSR1 and MerTK) on ercDCs in ccRCC tissue cell suspensions (*n* = 4). Overlay of MerTK^+^ cells on gated CD64A^+^CD204^+^ double-positive CD209^+^CD14^+^ ercDCs (upper right quadrant) depicted in pink. (**D**) Clustered heatmap of relative expression levels of genes associated with wound healing and tissue remodeling (**F**) and immunoregulation. (**D**,**F**) References are M2-macrophages and CD11c^LO^ macrophages from decidua (CD11c^LO^ MΦ_decidua). (**E**) Clustered heatmap depicting relative expression levels of genes associated with angiogenesis and invasion. Positive reference: in vitro-generated M2-macrophages; negative control: CD1c^+^ DCs from blood. (**E**, bottom) Surface expression of indicated proteins on ercDCs by flow cytometry of ccRCC tissue-cell suspensions and gating on ercDCs (CD209^+^CD14^+^ cells among CD45^+^ live single CD3^−^CD11c^+^ cells). (**G**) Confocal images of ccRCC tissues stained with CD209 (red), VSIG4 (green) and DAPI (blue). Original magnification, ×400. Cells co-expressing CD209 and VSIG4 are yellow. Images are merged fluorescent channels and maximal projection of 6–9 z-planes (z-step size = 0.7 µm). Top image: early stage ccRCC (RCC90, pT1aG2), bottom image: late stage ccRCC (RCC114, pT3cG2). (**H**) Markers associated with immunoinhibition. Exemplary histograms of ercDCs (CD209^+^CD14^+^ cells among gated CD45^+^CD11c^+^CD3^−^live single cells) from ccRCC tissue cell suspensions (*n* = 6) is shown. Histograms or dot plots are representative of at least 6 different tissue suspensions. Grey filled histogram is the isotype staining or FMO (fluorescence-minus-one), black line is the specific antibody. Numbers indicate the median of control-corrected median fluorescence intensity (delta MFI). Numbers indicate the control-corrected median fluorescence intensity.

**Figure 3 cells-11-03289-f003:**
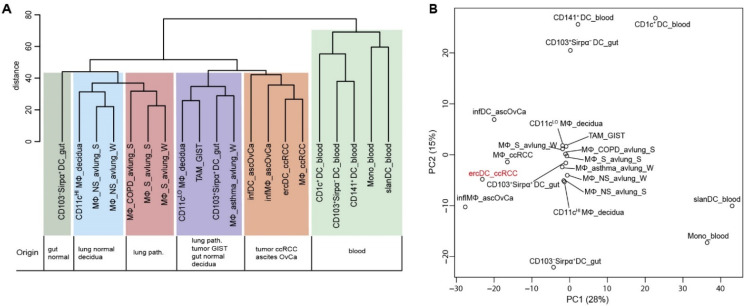
Transcriptome relationship of ercDC_ccRCC and myeloid cell types from blood and non-lymphoid healthy and pathological tissues. (**A**) Hierarchical clustering of indicated cell types based on median expression values of replicate samples considering only the top 50% of genes with highest variation across cell types (6107 “informative” genes). (**B**) PCA on same data set as described in (**A**). Shown are the first two PC that describe 28% and 15% of the variance, respectively.

**Figure 4 cells-11-03289-f004:**
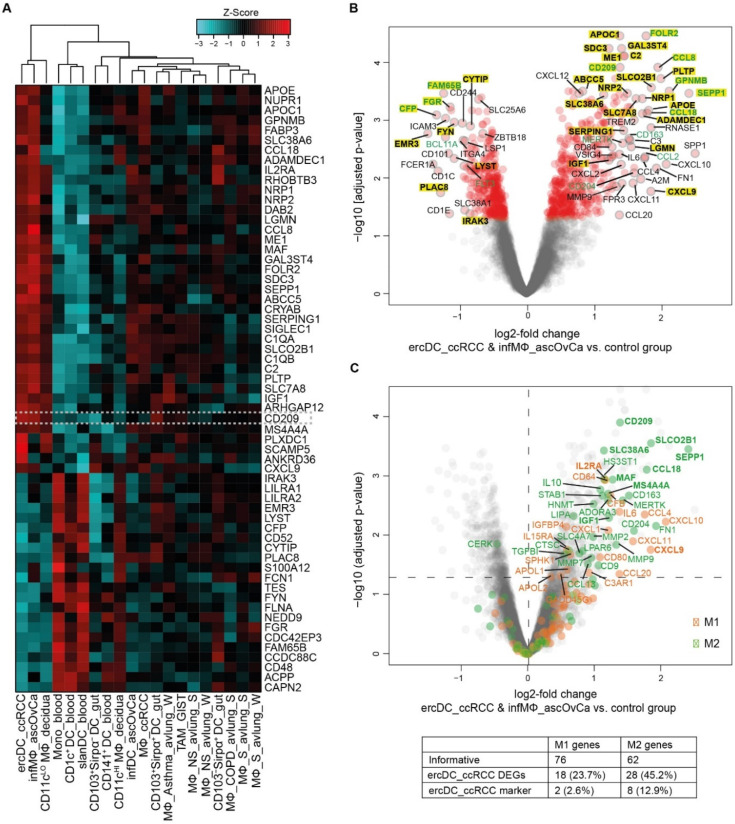
Marker gene profile and DEGs of ercDC_ccRCC. (**A**) Clustered heatmap of relative expression levels of 61 marker genes calculated with NSCM (see also Appendix A) comparing expression levels of ercDCs_ccRCC with those of the other cell types listed in Appendix A (control group). Dashed box highlights the gene CD209/DC-SIGN, which was used to distinguish ercDC_ccRCC from MΦ_ccRCC in flow cytometry. (**B**) Volcano plot depicting differences in gene expression across informative genes of ercDC_ccRCC&infMΦ_ascOvCa (*n* = 11) and control group (*n* = 177). DEGs are red symbols (adjusted *p* < 0.05). Symbols with assigned gene name have a grey edge. ErcDC_ccRCC marker genes are in bold and highlighted in yellow; green: genes discussed in the text. (**C**) Volcano plot depicting M1- and M2-macrophage-associated genes (orange and green); (grey): informative genes resulting from the comparison of ercDC_ccRCC&infMΦ_ascOvca and the control group. DEGs among informative genes are located above the dotted horizontal line (*p* < 0.05). The table shows the number and percentage of ercDC_ccRCC&infMΦ_ascOvCa DEGs and marker genes (bold) within the informative M1- and M2-macrophage-associated genes.

**Figure 5 cells-11-03289-f005:**
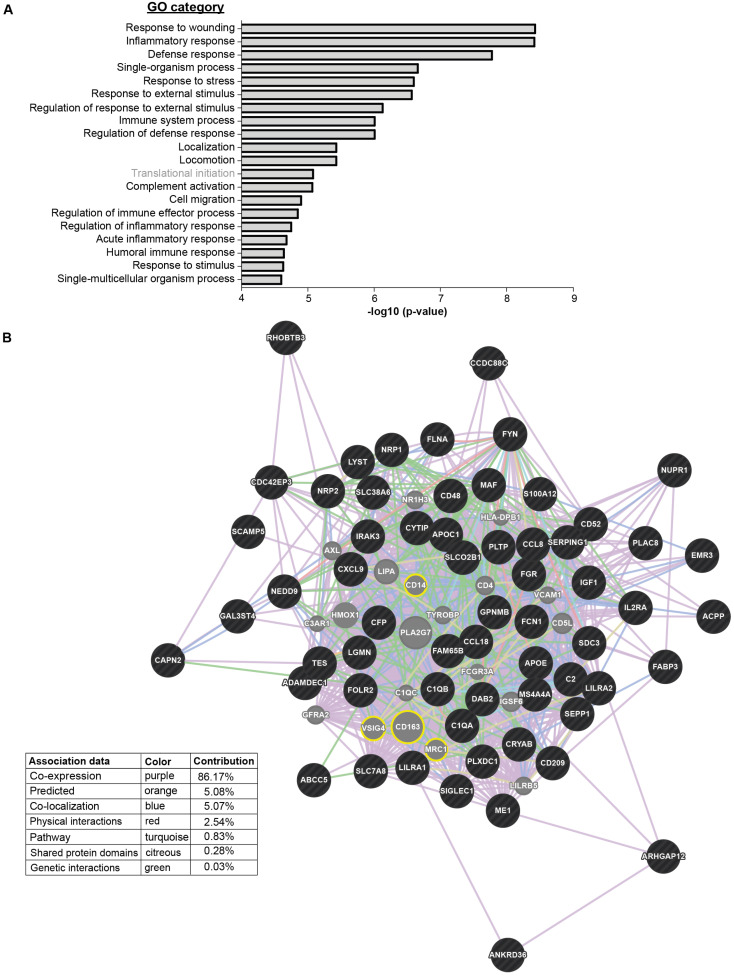
GO categories and functional network of ercDC_ccRCC. (**A**) Upregulated DEGs were enriched in 19 of the top 20 categories. Downregulated DEGs were only enriched in one category (“Translational initiation”) of the top 20 significantly enriched terms. Significance of GO category enrichment is depicted as −log10 (*p*-value). (**B**) Functional associations of ercDC_ccRCC marker genes (black) and their computed related genes (grey) by GeneMANIA network analysis. Yellow circles highlight related genes mentioned in the text. Associations were defined based on different criteria (colored lines, see Table).

**Figure 6 cells-11-03289-f006:**
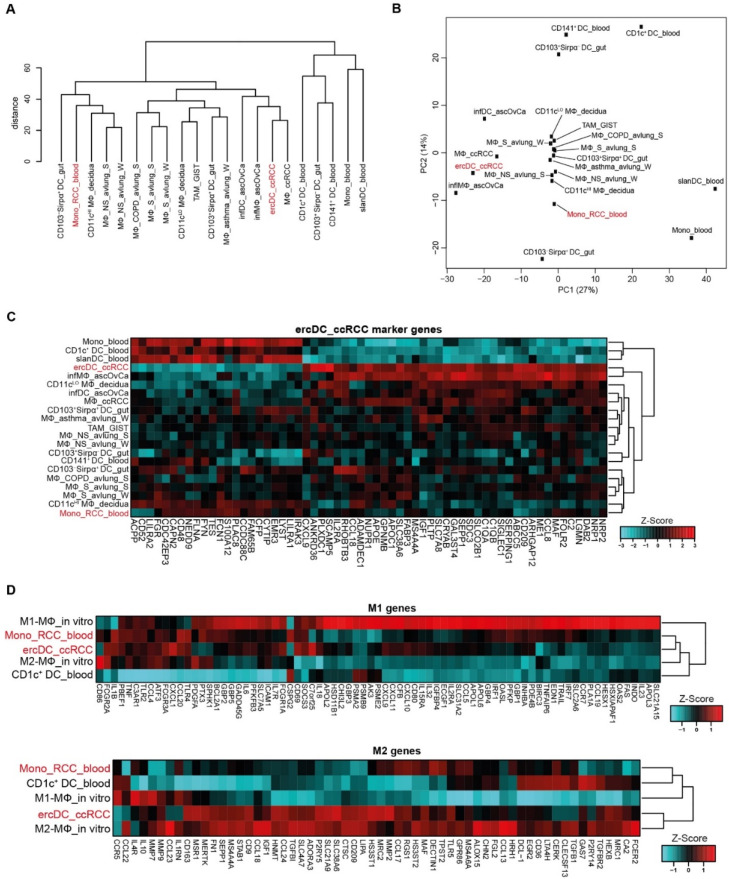
Transcriptome comparison of ercDC_ccRCC with published blood monocytes from RCC patients. All figures are based on expression levels with blood-specific genes not excluded. See Appendix A for results after exclusion of blood-specific genes. (**A**) Hierarchical clustering and (**B**) PCA of gene expression profiles. Analyses are based on informative genes for which the median expression value of replicates for each cell type was calculated. (**C**) Clustered heatmap of ercDC_ccRCC marker genes with blood monocytes of RCC patients (Mono_RCC_blood). (**D**) Clustered heatmaps showing relative expression levels of genes associated with M1- and M2-macrophages in Mono_RCC_blood and ercDC_ccRCC.

**Figure 7 cells-11-03289-f007:**
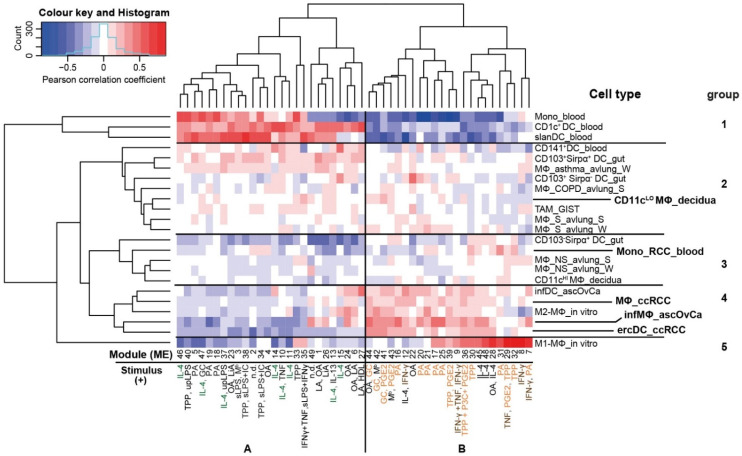
Polarization profile of ccRCC_ercDC reveals distinct tissue imprints. Heatmap of Pearson correlation coefficients calculated between module eigengenes (ME) and myeloid cell types. Cell types were clustered based on correlations with stimuli-associated eigengene modules. Depicted are the numbered modules with up to two significantly positively correlated stimuli for each module. Stimuli dominating in module group (**A**) are highlighted green, in module group (**B**) in orange and brown. Stimuli negatively correlating with the modules as well as the correlation rules for stimuli and modules are listed in Appendix A.

**Figure 8 cells-11-03289-f008:**
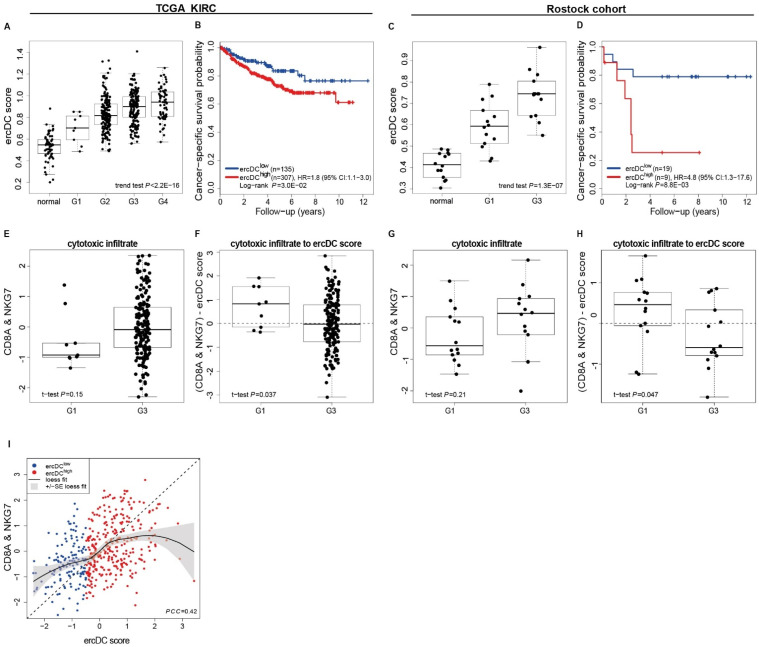
Association of ercDC_ccRCC marker gene profile with cancer-specific survival (CSS) and cytotoxic infiltrate. (**A**,**C**) Box plots of ercDC score values of TCGA KIRC (*n* = 439, no grading was available for 3 samples of TCGA KIRC, see Appendix A) and normal kidney (*n* = 65) or Rostock cohort (*n* = 14 for grade G1 and G3) (*p*-value by linear trend test). (**B**,**D**) Kaplan-Meier estimates of CSS of ccRCC tumors from TCGA KIRC (*n* = 442) and Rostock cohort (*n* = 28) predicted by the ercDC score. Partitioning into ercDC^low^ and ercDC^high^ risk groups using conditional inference trees with endpoint CSS. (**E**,**G**) Boxplots of the average of CD8A and NKG7 expression of tumors with grade G1 (*n* = 9) and G3 (*n* = 177) in TCGA KIRC or in Rostock cohort (each *n* = 14). Average expression values were standardized (z-scores). (**F**,**H**) Relationship between cytotoxic infiltrate and ercDC in tumors with grade G1 and G3 from TCGA KIRC or Rostock cohort. Average expression values and ercDC score were standardized (z-scores) before calculating the difference (**I**) Correlation of ercDC score with CD8A+NKG7 expression in TCGA KIRC (*n* = 442). A loess curve together with pointwise standard error displays the relationship between standardized average of CD8A and NKG7 expression and standardized ercDC score. Colors indicate the risk groups introduced in (**B**). PCC: Pearson correlation coefficient. Boxes refer to median and interquartile ranges with whiskers extending to a maximum of 1.5 times the interquartile range.

## Data Availability

Our data sets of human ercDCs and macrophages from ccRCC tissue, myeloid cells from PBMCs and in vitro-generated M1- and M2-macrophages are accessible via super series GSE108312. Transcriptome profiling data (“HTSeq-FPKM-UQ”) of the TCGA ccRCC cohorts (KIRC, LAML) were downloaded from the Genomic Data Commons Portal (https://gdc-portal.nci.nih.gov/ (accessed on 9 December 2016 and 15 December 2016, respectively)). Clinical data were obtained from the same platform on 10 October 2016 and 3 November 2016, respectively. Data are part of two doctoral theses at the medical faculty of the LMU.

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
