# Peer review of "Dendritic Cells or Macrophages? The Microenvironment of Human Clear Cell Renal Cell Carcinoma Imprints a Mosaic Myeloid Subtype Associated with Patient Survival"

_cells, 2022, doi:10.3390/cells11203289_

Round 1
Reviewer 1 Report
The authors have studied the gene expression pattern in CD14+CD209+ myeloid cells isolated from clear cell renal cell carcinoma patients and demonstrate that they mainly have a mixed M1/M2 macrophage gene expression pattern despite expression of both the CD14 monocyte and CD209 dendritic cell surface marker. They provide elegant bioinformatics analysis, including comparison to other normal and cancer-associated myeloid subsets, and inquiry regarding signaling pathways activated in this population that match those seen in RCC tumors (glucocorticoid, palmitic acid, PGE2). Flow cytometry confirms that this ercDC population is a unique uniform and confirms expression of surface markers identify by RNA analysis. They also analyze available data on 439 RCC patients to show that increased presence of this myeloid cell subset is associated with advanced tumor grade and worse survival.
Overall this a valuable addition to the literature on the phenotypes of human tumor-associated myeloid cells, characterizing a unique cell type with macrophage and DC characteristics.
1. In the introduction, can the authors clarify the proportion of ccRCC myeloid cells that the ercDC subset represents?
2. Line 249 states: CD1c+ DC and slanDCs were used as reference cell types for DC 269 characteristics. Can the authors clarify where this shown in the manuscript and further justify use of these subsets for comparison?
Author Response
Point by point Reply to Reviewer 1:
The authors have studied the gene expression pattern in CD14+CD209+ myeloid cells isolated from clear cell renal cell carcinoma patients and demonstrate that they mainly have a mixed M1/M2 macrophage gene expression pattern despite expression of both the CD14 monocyte and CD209 dendritic cell surface marker. They provide elegant bioinformatics analysis, including comparison to other normal and cancer-associated myeloid subsets, and inquiry regarding signaling pathways activated in this population that match those seen in RCC tumors (glucocorticoid, palmitic acid, PGE2). Flow cytometry confirms that this ercDC population is a unique uniform and confirms expression of surface markers identify by RNA analysis. They also analyze available data on 439 RCC patients to show that increased presence of this myeloid cell subset is associated with advanced tumor grade and worse survival.
Overall this a valuable addition to the literature on the phenotypes of human tumor-associated myeloid cells, characterizing a unique cell type with macrophage and DC characteristics.
- In the introduction, can the authors clarify the proportion of ccRCC myeloid cells that the ercDC subset represents?
Reply: In our previous work (Figel et al. ref 27), we had focused on CD209 expressing cells, which lead to the discovery of the ercDCs as a triple marker positive subset (CD209+CD14+CD163+). ErcDCs were found to represent the dominant fraction of all CD209+ cells in ccRCC (mean, 62%; range, 26% to 80%), whereas they were significantly less frequent in NKC (mean, 19%; range, 0% to 43%). The total number of myeloid cells was not determined.
In subsequent flow cytometry of tissue suspension, the total CD14+ cell subset was determined and within the CD14+ myeloid subset around 30% represented CD209+CD14+ ercDCs (range 10%-54%). This information is now included in the introduction, line 94-96:
“… due to its strong enrichment in the tumor center, where they represent over 60% of the CD209+ population (mean, 62%; range, 26% to 80%), compared to non-tumor kidney cortex (mean, 19%; range, 0% to 43%).“
- Line 249 states: CD1c+ DC and slanDCs were used as reference cell types for DC characteristics. Can the authors clarify where this shown in the manuscript and further justify use of these subsets for comparison?
Reply: slanDCs and CD1c+ DCs have described proinflammatory DC characteristics with IL‑12 production and T cell priming capacity [refs 50,51]. They were selected as reference representing cells with DC characteristics. We expected that they might be opposite to ercDC, for which a lack of IL-12 secretion and priming capacity had been observed (Figel et al., ref 27).
We added to the text (line 297 ff):
“CD1c+ DCs and slanDCs have been described to exhibit proinflammatory DC characteristics with IL-12 production and T cell priming capacity [refs 50,51]. They were used as DC reference cell types for in the comparison, to substitute for interstitial DCs, i.e. CD209+CD14- cells, which could not be sorted from ccRCC tissue cell suspensions due to low cell frequency. In the comparison to ercDCs they were expected to be opposite to the ercDCs, for which a lack of IL-12 secretion and priming capacity has been observed [27].
Line 405: “detailed information together with the rationale for their selecting are in supplemental procedures and table S2B”

Reviewer 2 Report
Brech D et al present a transcriptomic analysis to characterize a previously identified cell population in human clear cell renal carcinoma. The study is sound and the analyses are well done.
There are, however, a few issues that need to be addressed:
1) The study present a suite of transcriptomic analyses for the characterization of ercDCs. However, no functional analysis was done. Therefore, the hypothesis of a "functional analogy whereby ercDCs could contribute to T cell suppression etc..." (lines 94-98) was not really proven. If the authors present an hypothesis, this should be proven (or disproven). I suggest editing this part.
2) The analyses show that ercDCs belong to the macrophage lineage. They should then be renamed appropriately as it seems identifying them as DC is not applicable anymore.
3) In Figure 1A, the order of the cell types should be the same for the 2 heatmaps. This would help the reader.
4) Figure 2G should include a larger area to show better the cell composition of the tissue. Then zoomed-in panels could be used for details. Separation of color channels would further improve appreciation of the findings.
5) From Figure 2G, it seems earlier tumors have fewer ercDCs. This doesn't sit very well with the earlier hypothesis of ercDCs suppressing T cells. Only few ercDCs are unlikely to suppress immune response. However, ccRCC clearly progresses so there must be other mechanisms involved. Please clarify and discuss. This brings the issue back to an hypothesis that is not tested in the current manuscript.
6) Lines 551-553. The authors conclude that the two cell types have different tissue origin. Please explain better. The analysis is only based on gene expression, so it is not clear why the conclusion of different tissue origin. Can the authors demonstrate/disprove - for example - that these cells could have originated in the same tissue but then changed polarization do to the local tumor environment/milieu?
7) Lines 641-651. What are the common molecular features between ccRCC and ovarian cancer that could explain the presence of similar cell types? The authors indicate IL6, IL8, VEGF. However, it is unlikely that these factors are unique to only those cancer types. If those factors are expressed in other cancers, do these have ercDCs-like cells? From the data presented, this doesn't seem the case. Please provide additional evidence that could explain why these cell types are in ccRCC and ovarian cancer.
8) Lines 678-683. Again, I would be careful about mentioning functional inhibition of T cells by ercDCs. Before proposing a new targeting strategy, the authors should discuss what research needs to be done to confirm such interaction first. In addition - and assuming that ercDCs do indeed suppress immune response - from the (few) histological data, it seems like targeting the cells could be a strategy for advanced ccRCC, not so much for early treatment. A comment on this point would be good and interesting.
Author Response
Point by point Reply to Reviewer 2:
Brech D et al present a transcriptomic analysis to characterize a previously identified cell population in human clear cell renal carcinoma. The study is sound and the analyses are well done.
There are, however, a few issues that need to be addressed:
1) The study present a suite of transcriptomic analyses for the characterization of ercDCs. However, no functional analysis was done.
Therefore, the hypothesis of a "functional analogy whereby ercDCs could contribute to T cell suppression etc..." (lines 94-98) was not really proven. If the authors present an hypothesis, this should be proven (or disproven). I suggest editing this part.
Reply: It is correct that in the present manuscript no functional analysis were done since we focused on the description of the molecular characteristics of the ercDC cell subset. Previous experiments had provided evidence of immunomodulatory and tumor promoting attributes of ercDC, including the lack of IL-12 secretion and allopriming, promotion of tumor cell proliferation and curbed T cell recruitment due to the suppression of RANTES and CXCL10 secretion (Figel et al. 2010, and unpublished).
We have rephrased and extended the text in the introduction, line 105-111:
“Previous experiments suggested that ercDC were able to crosspresent antigen for T cell recognition. However, they did not secrete IL-12 and were unable to perform allogeneic T cell priming in vitro. Further evidence of immunomodulation was seen in co-culture with tumor cells and T cells where the presence of ercDCs enhanced tumor cell proliferation and reduced CD8 T cell recruitment. Immune histology supported this evidence as lower numbers of CD8+ and NK cells were seen in ccRCC tissues with high ercDC content [27]).”
2) The analyses show that ercDCs belong to the macrophage lineage. They should then be renamed appropriately as it seems identifying them as DC is not applicable anymore.
Reply: We were considering the name change for some time. CD209+ and CD209+CD14+ cells are sometimes called DCs other times macrophages in the literature (Kammerer-U et al. 2003; Tailleux-L et al. 2005; Soilleux-EJ et al. 2002; Krutzik-SR et al. 2005). We selected the DC assignment based on the large literature of murine kidney myeloid cells, wherein parenchymal myeloid cells co-expressing macrophage and DC markers were assigned to “dendritic cells” (see refs 80: Nelson et al., ref 140: Gottschalk et al.; ref 29: Scholz et al.; ref 67: Kurts et al.; ref 111: Ferenbach et al.).
Considering the reviewer’s comment and the transcriptome data, we decided to change the name to ercMP in future reference. A respective text is included in the conclusion, line 772:
“Based on this now available information, we suggest changing the name to ercMP (enriched in renal cell carcinoma macrophages) in future reference.”).
3) In Figure 1A, the order of the cell types should be the same for the 2 heatmaps. This would help the reader.
Reply: The figure 1A has been changed accordingly.
4) Figure 2G should include a larger area to show better the cell composition of the tissue. Then zoomed-in panels could be used for details. Separation of color channels would further improve appreciation of the findings.
Reply: A larger area has been selected and is shown in the revised manuscript, Figure 2G Line 370.
5) From Figure 2G, it seems earlier tumors have fewer ercDCs. This doesn't sit very well with the earlier hypothesis of ercDCs suppressing T cells. Only few ercDCs are unlikely to suppress immune response. However, ccRCC clearly progresses so there must be other mechanisms involved. Please clarify and discuss. This brings the issue back to an hypothesis [… T cell suppression] that is not tested in the current manuscript.
Reply: We appreciate the reviewer’s thoughts and changed and extended the text avoiding “T cell inhibition/suppression”, replacing it with “immunomodulation or immune escape” where appropriate.
As a remark to the reviewer regarding low cell numbers and immunosuppression: we believe that these are not excluding features, as an immune suppression through Tregs is well documented in spite their low numbers.
We agree with the reviewer, that earlier tumors have fewer ercDCs. This was seen in previous work judged by TNM stage (Figel et al.) and is recapitulated in the present manuscript judged by the tumor grade (Figure 8). In the current manuscript, we did not present data of immune cell suppression because the focus was set on describing the molecular features of the ercDCs. We edited the manuscript referring to previous work in the introduction line, 105-111:
“Previous experiments suggested that ercDC were able to crosspresent antigen for T cell recognition. However, they did not secrete IL-12 and were unable to perform allogeneic T cell priming in vitro. Further evidence of immunomodulation was seen in co-culture with tumor cells and T cells where the presence of ercDCs enhanced tumor cell proliferation and reduced CD8 T cell recruitment. Immune histology supported this evidence as lower numbers of CD8+ and NK cells were seen in ccRCC tissues with high ercDC content [27].”
Additionally, we added new panels to Figure 8 (E-I), which show that the content of the ercDC is associated with changes in the immune composition of the tissue. This supports our previous work where a similar change was observed based on counted cell numbers in histology. The included text is in lines 605 ff.
“Early stage tumors (G1) as compared to later stage tumors (G3) not only showed a lower ercDC score, but by trend also had a lower cytotoxic infiltrate, approximated by the mean expression levels of CD8A and NKG7 (natural killer cell granule protein 7 ex-pressed by NK cells and CD8 T cells in ccRCC [22]) (figure 8E, G).
Correlation analysis of ercDC score and cytotoxic infiltrate highlights a parallel ex-pansion of cytotoxic infiltrate and ercDC score (Pearson correlation coefficient = 0.42). This relationship, however, was not proportional as the loess curve fitted using the standardized values was above the diagonal at lower ercDC values and below the diag-onal at higher ercDC values (figure 8I). Thus, the cytotoxic infiltrate dominated the ercDC score in earlier stage tumors while the ercDC score offset the cytotoxic infiltrate in the later stage tumors, as seen also by the calculated ratios (figure 8F, H). This relation-ship may explain, in part, the finding that a larger CD8 T cell infiltrate is not beneficially associated with survival in ccRCC, contrasting most other tumors [96].
The relationship is consistent with our previous histology work where a propor-tionally lower CD8 cell count was found in tumors with high ercDC numbers [27]. While other reasons may cause this relationship, ercDCs may actively contribute by curbing the cytotoxic immune cell recruitment through reducing Th1 recruiting chemokines previ-ously observed in in vitro T cell/tumor cell co-cultures.
The more favorable composition of the infiltrate in G1 tumors might contribute to the prognostically better outcome of patients with early stage tumors. In G1 less ercDCs may be sufficient for immuninhibition in the context of a lower cytotoxic infiltrate and ensuing interaction between T cells, macrophages and tumor cells can lead to tumor progression through macrophage-secreted tumor promoting factors, such as CCL8, CCL18, and MMP9, which are part of the ercDC signature.”
Regarding a role of ercDCs in tumor progression, we added to the discussion, lines 740 ff:
“While there are certainly additional mechanisms related to tumor progression and therapy resistance, like insufficient CD8 or NK cell counts or T cell dysfunction or inhibiting tissue factors, such as lactic acidosis [27,130-132], the finding by Braun et al. provides support for a role of macrophages as a mechanism of immune therapy resistance [102]. Observed contacts between macrophages and exhausted T cells suggest a T cell inhibitory communication that may be mediated by surface molecules. In this respect, our re-port of a novel series of proteins expressed on ercDCs is of interest. Especially, markers such as VSIG4 [64,133-135] and GPNMB [136,137], are discussed in the literature in the context of T cell inhibition and cell cycle arrest. While PD-L1, PD-L2 and TIM-3 were only marginally expressed by ercDCs, these new markers represent promising targets to moderate the communication that ercDC might mediate through T cell contact. NRP1 [138] and CSF-1R are additional possible targets expressed by ercDCs, for which therapeutic reagents are currently in clinical studies (https://clinicaltrials.gov/). The relevance of these markers for functional alteration of a T cell antitumor immune response will need to be assessed in experimental settings. Knockdown of these newly identified markers may reveal if ercDCs can be modulated to participate in productive antitumor immune response. Combined with checkpoint blockade or other therapeutic strategies, targeting these proteins may improve treatment outcome, especially for tumors with a high ercDC score.”
6) Lines 551-553. The authors conclude that the two cell types have different tissue origin. Please explain better. The analysis is only based on gene expression, so it is not clear why the conclusion of different tissue origin. Can the authors demonstrate/disprove - for example - that these cells could have originated in the same tissue but then changed polarization due to the local tumor environment/milieu?
Reply: We appreciate the reviewer’s careful reading of the manuscript and the insightful comment. We apologize for choosing the wrong word. We did not study the tissue origin (likely, both cell types originate from BM myeloid cells), but we addressed features that occurred due to the exposure to different local milieus. We changed the text accordingly, line 573 f:
“…likely reflecting their exposure to different local milieus.”
7) Lines 641-651. What are the common molecular features between ccRCC and ovarian cancer that could explain the presence of similar cell types? The authors indicate IL6, IL8, VEGF. However, it is unlikely that these factors are unique to only those cancer types. If those factors are expressed in other cancers, do these have ercDCs-like cells? From the data presented, this doesn't seem the case. Please provide additional evidence that could explain why these cell types are in ccRCC and ovarian cancer.
Reply: IL-6, IL-8 and VEGF were found to be mandatory for polarization of monocytes to generate a proxy of ercDC in vitro (Figel et al., ref 27). These factors are not unique to RCC and ovarian cancer, and, we agree with the reviewer, that myeloid cells with similar marker expression are likely not unique to the RCC tumor. In that regard, we have reported previously that CD209+CD14+ ercDC were found in the non-tumor (healthy) kidney cortex (Figel et al.), albeit at much lower frequency; therefore the proposed name enriched in renal cell carcinoma (Figel et al., ref 27). We included this information in the introduction, line 93ff:
“We designated this myeloid subpopulation “enriched-in-renal-carcinoma DCs” (ercDCs), due to its strong enrichment in the tumor center, where they represent over 60% of the CD209+ population (mean, 62%; range, 26% to 80%), compared to non-tumor kidney cortex (mean, 19%; range, 0% to 43%).”
In addition, we have reported previously the presence of cells with similar marker expression in inflammatory kidney disease (ref 110: Segerer et al.). Newly available transcriptome data of lupus nephritis support this finding, discussed in lines 673. Furthermore, triple marker positive cells have been described in dermis and the decidua. The revised manuscript has new references and discussion related to these findings, line 667-672:
„A relationship between the ercDC from RCC tumor tissue and myeloid cells from chronic inflammatory kidney pathologies was previously suggested based on the triple marker staining of CD14, CD209 and CD163 [27,110]. CD14+CD209+CD163+ triple positive cells are also described in the human dermis, in a murine leprosy model, and in the decidua of early human pregnancy [27,111-113]. The presence of these cells was linked to deviated immune responses, allowing bacterial or embryo persistence. As this seemed to parallel the situation of tumors, available transcript data were compared to the ercDC transcriptome. However, despite the conceptual similarity, the myeloid cells from these tissues did not appear to be closely related to our ercDCs. Yet, the recently available single cell sequencing data describing the immune landscape of lupus nephritis revealed five clusters of myeloid cells [114], whereby cells in… “
Furthermore, we discuss the observed close relationship to inflammatory macrophages of ovarian cancer ascites in more detail, lines 705ff:
„The observed close relationship of ercDCs to the infMΦ_ascOvCa may be explained in part as both are myeloid cells from epithelial tumors. The TAM_GIST are from a mesenchymal tumor, and the dissimilar transcriptional data might result from the different tissue milieu. It is, however, interesting that of the two myeloid cell types from the ovarian cancer (infMΦ_ascOvCa and infDC_ascOvCa), only the infMΦ_ascOvCa was computationally associated with ercDCs, showing stronger relationship than the MΦ_ccRCC. This suggests that within the tumor microenvironments different niches exist which shape the transcriptional profile of the residing myeloid cells to become dissimilar to an extent that they more closely match to cells in a different tissue. Since our analysis was performed, additional transcriptomes of human myeloid cells from non-lymphoid tissue have become available, including those of human breast [115,127] and hepatocellular carcinomas [128]. ErcDC share marker expression with these macrophages, including FOLR2, CCL8, MMP2 and MMP9 as well as APOE, SEPP1. It will be exciting to learn if the closeness seen with the infMΦ_ascOvCa can be expanded to other macrophage populations as the field develops and we learn more about the biology at work. As an overarching conclusion of current publications, our data of ercDC transcriptome confirms the notion that the profile of tissue macrophages is very complex integrating markers of various polarization states.”
8) Lines 678-683. Again, I would be careful about mentioning functional inhibition of T cells by ercDCs. Before proposing a new targeting strategy, the authors should discuss what research needs to be done to confirm such interaction first.
In addition - and assuming that ercDCs do indeed suppress immune response - from the (few) histological data, it seems like targeting the cells could be a strategy for advanced ccRCC, not so much for early treatment. A comment on this point would be good and interesting.
Reply: We appreciate the reviewer’s remark and changed the text avoiding “T cell inhibition/suppression” where possible.
A comment to the point that targeting ercDCs could be a strategy for a subset of patients, i.e. those who have tumors with high ercDCs, is included in the discussion, 756 ff:
“Combined with checkpoint blockade or other therapeutic strategies, targeting these proteins may improve treatment outcome, especially for tumors with a high ercDC score”) and the conclusion (line 778 ff: “Targeting ercDCs in combination with immunotherapy, particularly in situations, where tumors have high ercDC content, may expand the range of patients that can be effectively treated with immunotherapy.”).
Additionally we expanded the text to highlight that future work is required to evaluate if the new markers as well as the ercDCs have therapeutic potential, line 753ff:
“The relevance of these markers for functional alteration of a T cell antitumor immune response will need to be assessed in experimental settings. Knockdown of these newly identified markers may reveal if ercDCs can be modulated to participate in productive antitumor immune response.”)
and lines 776ff:
“As a correlation may not necessarily equate to causality, the potential role that ercDCs might play in the regulation of the antitumor response requires testing in experimental models.”

Round 2
Reviewer 2 Report
I thank the authors for replying to my comments. I believe the manuscript has been improved significantly.